# Hydrogen and CNT Production by Methane Cracking Using Ni–Cu and Co–Cu Catalysts Supported on Argan-Derived Carbon

Fernando Cazaña [1], Zainab Afailal [2], Miguel González-Martín [1], José Luis Sánchez [2], Nieves Latorre [1], Eva Romeo [1], Jesús Arauzo [2] and Antonio Monzón [1,*]

1   Departamento de Ingeniería Química y Tecnologías del Medio Ambiente, Instituto de Nanociencia y Materiales de Aragón, CSIC-Universidad de Zaragoza, 50018 Zaragoza, Spain; fcazana@unizar.es (F.C.); m.gonzalezmartin@posta.unizar.es (M.G.-M.); nlatorre@unizar.es (N.L.); evaromeo@unizar.es (E.R.)
2   Departamento de Ingeniería Química y Tecnologías del Medio Ambiente, Instituto de Ingeniería de Aragón (I3A), Universidad de Zaragoza, 50018 Zaragoza, Spain; zainabafailal@unizar.es (Z.A.); jlsance@unizar.es (J.L.S.); jarauzo@unizar.es (J.A.)
*   Correspondence: amonzon@unizar.es

**Abstract:** The 21st century arrived with global growth of energy demand caused by population and standard of living increases. In this context, a suitable alternative to produce COx-free $H_2$ is the catalytic decomposition of methane (CDM), which also allows for obtaining high-value-added carbonaceous nanomaterials (CNMs), such as carbon nanotubes (CNTs). This work presents the results obtained in the co-production of $CO_x$-free hydrogen and CNTs by CDM using Ni–Cu and Co–Cu catalysts supported on carbon derived from Argan (Argania spinosa) shell (ArDC). The results show that the operation at 900 °C and a feed-ratio $CH_4$:$H_2$ = 2 with the Ni–Cu/ArDC catalyst is the most active, producing 3.7 $g_C$/$g_{metal}$ after 2 h of reaction (equivalent to average hydrogen productivity of 0.61 g $H_2$/$g_{metal}$·h). The lower productivity of the Co–Cu/ArDC catalyst (1.4 $g_C$/$g_{metal}$) could be caused by the higher proportion of small metallic NPs (<5 nm) that remain confined inside the micropores of the carbonaceous support, hindering the formation and growth of the CNTs. The TEM and Raman results indicate that the Co–Cu catalyst is able to selectively produce CNTs of high quality at temperatures below 850 °C, attaining the best results at 800 °C. The results obtained in this work also show the elevated potential of Argan residues, as a representative of other lignocellulosic raw materials, in the development of carbonaceous materials and nanomaterials of high added-value.

**Keywords:** $CO_x$-free hydrogen; CNTs; CDM; methane; argan-derived carbon; Ni–Cu; Co–Cu

## 1. Introduction

The potential applications of the carbonaceous nanomaterials (CNMs), for example, carbon nanotubes (CNTs), carbon nanofibers (CNFs) or graphene, due to their unique mechanical, electronic, chemical and physical properties, have motivated the enormous research effort carried out during the last few decades in almost all fields of nanoscience and nanotechnology [1–8].

Among the currently available technologies for the production of CNTs, the catalytic decomposition of light hydrocarbons in the gas (or vapor) phase stands as the most interesting method to satisfy the large potential demand, due to its easy scalability for production at large-scale and low-cost [9,10]. If the carbon source used is methane, the co-production of pure hydrogen is an additional and very relevant advantage due to the increasing necessity of the $CO_x$-free hydrogen in the actual energetic, environmental and political scenario [11–14]. Significant efforts were made to find the most suitable operating conditions and catalyst compositions to provide both, high carbon and hydrogen yields, and the necessary selectivity and quality for the desired carbon structure. However, the

application of this process is still limited because of the rapid deactivation of the catalysts commonly employed [13,15,16].

Several metallic and also non-metallic (generally carbon materials) components of the catalysts were tested in the reaction of methane cracking. Thus, catalysts based on transition metals such as Ni, Fe, and Co are the most active, can operate at moderate temperature, and are also able to produce valuable CNMs as co-product [17–22]. The incorporation of a second metal as a catalytic promoter was reported to have positive effects on the activity and stability [23–26]. In previous studies, we demonstrated that an appropriate design of the catalyst composition and an adequate selection of the operating conditions (mainly temperature and $CH_4/H_2$ feed-ratio) allows not only the control of hydrogen yield in the catalytic decomposition of methane (CDM) reaction but also of the selectivity towards the desired carbonaceous nanomaterials [27]. Thus, reaction temperatures below 800–850 °C favor the formation of CNTs [27], while temperatures above 900–950 °C are more prone to produce graphene-related materials (GRMs) [24,28], depending on the catalyst composition.

Moreover, the increasing concern for the environment is leading to the study of new processes to obtain high added-value products and materials using renewable natural bioresources [11,29–31]. In this concern, one of the natural resources that can be used is the shells of argan (Argania Spinosa), a waste coming from the use of the kernels of their nuts for the production of argan oil, one of the most expensive vegetable oils in the world. The argan fruits are principally produced in Morocco and Tunisia, with an estimated production of 80,000 tons per year [32].

In this regard, the use of vine shoots wastes and cellulose as raw materials for the preparation of carbon-based supports of metallic catalysts is being investigated in our group [24,30,33,34]. These catalysts, prepared by controlled thermal decomposition of the impregnated raw materials (e.g., argan shells) with the metallic precursors exhibited high catalytic performance due to the good dispersion of metallic nanoparticles and the controlled textural properties of the support [35–37]. This technique easily allows converting hierarchical structures formed by a biological process into inorganic materials with high potential for different applications due to their porous texture developed during the preparation [38–42]. In addition, as the natural raw resource is previously impregnated with metallic catalytic precursors, the catalyst is obtained in one step [24]. The possibility of using different raw materials and metals makes this method a very useful and versatile tool to prepare mono or multimetallic supported catalysts with a wide range of compositions [24,35,37].

Thus, the ultimate goal is to demonstrate that it is possible to prepare proper catalytic supports using lignocellulosic residua ("Biomass Derived Carbons—BDC"), without very low commercial value, such as the "Argan Derived Carbon—ArDC" used here. The objective is to use these sustainable supports as an alternative to the traditional metallic oxides, such as silica or alumina, which have a large environmental and economic impact. In this context, we present here the results of the use of Ni–Cu and Co–Cu catalysts supported by Argan-Derived Carbon (ArDC) on the reaction of catalytic decomposition of methane to produce $CO_x$-free hydrogen and carbon nanomaterials.

The composition of the active phases (Ni–Cu and Co–Cu) was chosen based on previous works with the aim of obtaining different materials such as carbon nanotubes [27], or even graphene-related materials (GRMs) [24]. As a means to optimize the productivity and also the quality of the carbonaceous material formed, the effect of the main operating conditions during the reaction, i.e., feed composition and reaction temperature, was investigated.

## 2. Materials and Methods

### 2.1. Catalysts Preparation

Ni–Cu and Co–Cu/ArDC catalysts were prepared by thermal decomposition under reductive atmosphere of milled Argan shells, as described elsewhere [36]. First, the dried Argan shell was impregnated by incipient wetness with the appropriate amount of an

aqueous solution containing the precursor salts ($Ni(NO_3)_2 \cdot 6H_2O$ provided by Alfa Aesar, $Co(NO_3)_2 \cdot 6H_2O$ and $Cu(NO_3)_2 \cdot 3H_2O$ provided by Sigma-Aldrich), to obtain a nominal weight composition of 5%Ni−1.35%Cu (atomic ratio Ni/Cu = 4) and 5%Co−1.35%Cu (atomic ratio Co/Cu = 4) with respect to the initial amount of Argan shell. After impregnation, the solid was dried at 100 °C overnight under 100 mL/min $N_2$ and thermally decomposed at 800 °C under reductive atmosphere (15% $H_2$/85% $N_2$) for 75 min with a heating rate of 50 °C/min. Finally, the catalyst was milled and sieved obtaining a particle size distribution ranging from 80 μm to 200 μm. These values are selected to avoid any internal diffusion limitations. The presence of the external limitations is also minimized by using high flow rates (700 mL/min, equivalent to 1680 mL/gcat.h).

### 2.2. Catalytic Decomposition of Methane

The catalytic decomposition of methane (CDM) was used to obtain carbon nanotubes (CNTs) at atmospheric pressure in a quartz thermobalance (CI Precision Ltd., Salisbury, UK, model MK2, https://www.ciprecision.com/ accessed on 20 February 2022) operated as a continuous fixed-bed differential reactor. In order to ensure the integrity of the equipment, and avoid any corrosive damage to the head of the balance, a continuous flow of inert gas was used, $N_2$ in our case. On the other part, the objective of introducing hydrogen into the feed mixture is to investigate its effect on the activity and stability of the catalysts during the reaction. In absence of $H_2$, the deactivation of the catalyst is very rapid.

Carbon mass evolution and temperature reaction were usually recorded during 120 min of reaction. The temperature range studied was 750–950 °C while the feed gas composition was varied from 0.5 to 3 of $CH_4$:$H_2$ ratio, using $N_2$ as balance gas (total flow rate 700 mL/min). In a characteristic experiment, 25 mg of the catalyst was placed into a copper mesh sample holder and then, the sample was heated at 10 °C/min under $N_2$ flow (700 mL/min) until reaching the reaction temperature. Once the desired reaction temperature was achieved, the reactive gas mixture $CH_4$/$H_2$/$N_2$ (700 mL/min) was fed into the reactor keeping constant the temperature during the reaction. Finally, the sample was cooled down to room temperature under $N_2$.

### 2.3. Catalysts and Carbonaceous Nanomaterials Characterization

The metal content of both catalysts was calculated by thermogravimetric analysis under oxidative atmosphere (air, 50 mL/min) in a Mettler Toledo TGA/SDTA 851 equipment. The desired amount of catalyst (~10 mg) was heated from 35 °C to 1000 °C with a heating rate of 10 °C/min. The final percentage of Ni–Cu and Co–Cu was determined considering the nominal metal quantity incorporated and the resulting mixture of metal oxides obtained upon combustion of the ArDC support during the TGA-air experiment.

Specific surface area and porosity values for both catalysts were obtained from $N_2$ adsorption–desorption isotherms measured at −196 °C using a TriStar 3000 instrument. Previous to the analysis, the catalysts were degassed at 200 °C for 8 h. BET specific surface area was calculated in the relative pressure range of $P/P_0$ = 0.01–0.10. Total pore volume was obtained at the maximum relative pressure reached by the adsorption branch ($P/P_0 > 0.985$), while the micropore volume was estimated by the t-plot method. Crystalline phase identification of the fresh catalysts was performed by X-ray diffraction (XRD) in a Rigaku D/Max 2500 apparatus from 5° to 90° 2θ degrees using Cu Kα radiation ($\lambda$ = 1.5406 Å).

Morphological and structural information on the carbonaceous nanomaterials grown was obtained by electron microscopy. Transmission electron microscopy (TEM) images were acquired using an FEI Tecnai T-20 microscope operated at 200 kV. Scanning electron microscopy (SEM) and energy-dispersive X-ray spectroscopy (EDS) analysis were carried out in an FEI Inspect F50 microscope.

The carbonaceous structure of the Argan-derived carbon support and the quality of the CNTs obtained were characterized by Raman spectroscopy in a WiTec Alpha300 confocal Raman microscope using a 532 nm laser excitation beam. The intensity ratios $I_D/I_G$, $I_{2D}/I_G$

of the characteristic D (~1350 cm$^{-1}$), G (~1580 cm$^{-1}$), and 2D (~2690 cm$^{-1}$) bands from different sample spots (5 spectra) were used to evaluate the defects in the structure of the carbonaceous nanomaterials obtained.

## 3. Results

### 3.1. Fresh Catalyst Characterization

The Ni–Cu/ArDC and Co–Cu/ArDC catalysts were synthesized with nominal contents (wt%) of 5% Ni and 1.35% Cu and 5% Co and 1.35% Cu, respectively, with respect to the initial amount of milled Argan shell. After synthesis, the amounts of metal, calculated from the TGA-air data, were ca. Ni(16%)–Cu(4%) and Co(18%)–Cu(5%). This increment in the metal content is caused by the weight loss of carbonaceous support during the thermal decomposition of the natural source, a decisive stage to control the dispersion and the final content of the metal nanoparticles in the synthesized catalysts [24].

Information about the crystalline phase of the active metals before the reaction was addressed by the XRD technique in Figure 1. For the Ni–Cu/ArDC catalyst, it is observed that the peaks appearing at 44.3°, 51.6° and 76.0° do not correspond to the metallic Ni (45-1027 JCPDS) or Cu (04–0806 JCPDS) patterns. The shift observed in the 2θ value for these peaks can be associated with the formation of a Ni–Cu alloy [43]. However, the XRD pattern depicted in Figure 1 for Co–Cu/ArDC shows that both Co and Cu were present in metallic form (15–0806 JCPDS for Co$^0$ and 04–0806 JCPDS for Cu$^0$), not observing the formation of an alloy in this case. In addition, the Ni–Cu/ArDC and Co–Cu/ArDC patterns do not show any peak associated with the presence of oxidized species, confirming that the catalysts are completely in the reduced state.

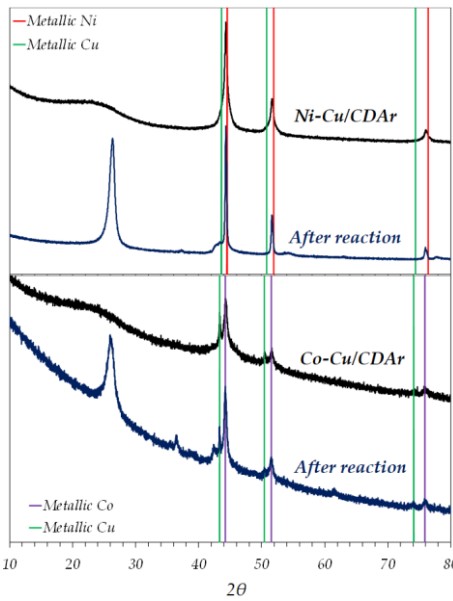

**Figure 1.** XRD patterns of fresh and used catalysts (800 °C; CH$_4$/H$_2$/N$_2$(%): 28.6/14.3/57.1).

The wide diffraction peak at about 2θ = 26°, attributed to the (002) plane of the hexagonal graphite structure (75–2078 JCPDS), confirms the amorphous nature of the carbonaceous ArDC support.

As it is shown in Figure 1, XRD results indicate that the metallic phases of both catalysts (Ni–Cu and Co–Cu) do not suffer relevant modifications during the reaction. The new peaks at 2θ = 26° correspond to the CNMs formed in each case, being more intense in the Ni–Cu sample, in agreement with the Raman and TEM results presented in the next paragraph.

Morphology of the bulk Ni–Cu/ArDC and Co–Cu/ArDC catalysts was addressed by electron microscopy images presented in Figure 2. The SEM images show the macrostruc-

ture and smoothness characteristic of this type of carbonized material for both catalysts prepared. In addition, TEM images obtained for the two fresh catalysts indicate that the metal nanoparticles were well distributed on the ArDC support owing to their characteristic biomorphic textural structure. TEM images were employed to estimate the particle size of the incorporated metals by measuring not less than 500 particles from different sample areas. The Ni–Cu/ArDC sample shows a trimodal metal particle size distribution, finding relatively small particles (~7 nm), intermediate particles (~17 nm) and large particles (~35 nm). Moreover, the Co–Cu/ArDC catalyst shows a bimodal metal particle size distribution with a large number of spherical nanoparticles about 5 nm and a low number of nanoparticles about 31 nm. In both cases, these metal size distributions are a consequence of the heating rate and high temperature used during the catalyst synthesis [36].

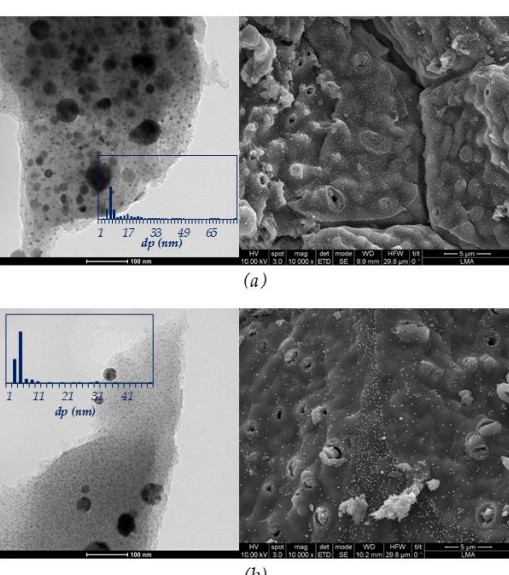

**Figure 2.** Electron microscopy study of the fresh catalysts: (**a**) Ni–Cu/ArDC (**b**) Co–Cu/ArDC.

Table 1 summarizes the textural properties of the fresh catalysts before reaction. In this Table 1, the values corresponding to the same type of metallic catalysts (Ni–Cu and Co–Cu), but supported on other carbon-based material, carbon derived from cellulose (CDC), and prepared using the same method are also included [24,27]. The main difference was found in the higher microporosity of the catalysts supported by the argan-derived carbon, especially in the case of the Ni–Cu catalyst. In this case, the Ni–Cu/ArDC has a 63% less pore volume but its microporosity is 2.5 times higher than the Co–Cu/ArDC one. Comparing the two catalysts supported by ArDC, both present similar textural properties corresponding to highly microporous materials, although the Ni-based sample has a slightly less developed porosity.

**Table 1.** Textural properties of the fresh catalysts.

| Sample | BET Area (m$^2$/g) | Pore Vol. [1] (cm$^3$/g) | μpore Vol. [2] (cm$^3$/g) | μpore Vol. (%) |
|---|---|---|---|---|
| Ni–Cu/ArDC | 404 | 0.168 | 0.138 | 82 |
| Co–Cu/ArDC | 433 | 0.182 | 0.164 | 90 |
| Ni–Cu/CDC [24] | 343 | 0.451 | 0.148 | 33 |
| Co–Cu/CDC [27] | 438 | 0.206 | 0.160 | 78 |

[1] Total pore volume at $P/P_0 > 0.989$. [2] Estimated with the t-plot method.

The carbonaceous structure of the fresh catalysts was also characterized by Raman spectroscopy shown in Figure 4. For both catalysts, the characteristic G band attributed to the in-plane vibration of the C sp$^2$-hybridized bonds was observed at ~1590 cm$^{-1}$, while

the D band was associated with defects within the C sp$^2$ network of the carbonaceous support was detected at ~1350 cm$^{-1}$ [44]. The wideness of these peaks (D and G bands) is a consequence of the different structural contributions of the several kinds of graphitic domains created during the thermal decomposition stage [36,45]. The wide and weak peak at about 2700 cm$^{-1}$, known as the overtone of the D band, indicates the amorphous nature of the catalyst support, which is in agreement with the XRD results shown in Figure 1.

### 3.2. Catalytic Decomposition of Methane

### 3.2.1. Influence of Reaction Temperature

The effect of the reaction temperature on the activity, productivity and morphology of the CNMs grown on both catalysts was evaluated in the interval from 750 °C to 950 °C, using a CH$_4$/H$_2$ ratio equal to 2 (28.6%CH$_4$/14.3%H$_2$/57.1%N$_2$). Figure 3 shows the evolution along time of the carbon concentration, $m_C$ (g$_C$/g$_{metal}$), at different reaction temperatures for Ni–Cu/ArDC (Figure 3a) and Co–Cu/ArDC (Figure 3b). In addition, the summary of the results of activity and productivity after 2 h of reaction time for both catalysts is presented in Table 2.

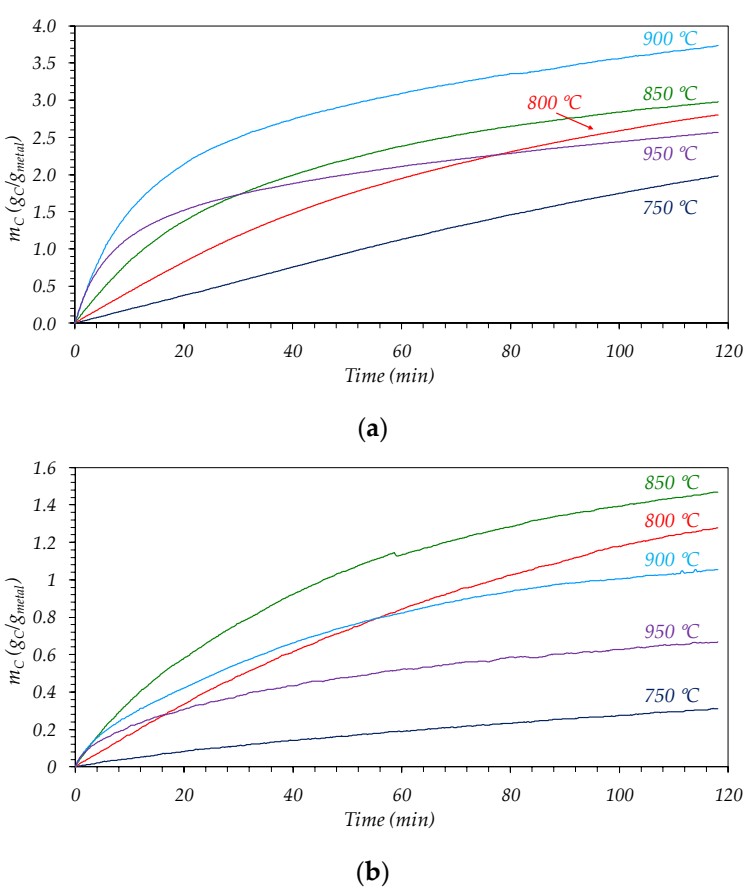

**Figure 3.** *Cont.*

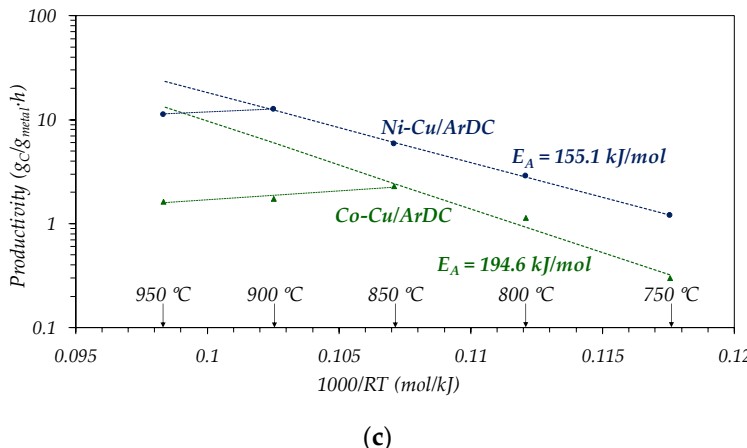

**(c)**

**Figure 3.** Influence of reaction temperature on evolution of carbon concentration, $m_C$: (**a**) Ni–Cu/ArDC; (**b**) Co–Cu/ArDC; (**c**) Final productivity at 2 h (Dashed line: Arrhenius plot).

**Table 2.** Influence of temperature on the initial reaction rate and carbon productivity *.

| Temperature (°C) | Ni–Cu/ArDC | | | Co–Cu/ArDC | | |
|---|---|---|---|---|---|---|
| | $r_{C0}$ ($g_C/g_{metal}\cdot min$) | Carbon Product. ($g_C/g_{metal}\cdot h$) * | Carbon Product./$r_{C0}$ | $r_{C0}$ ($g_C/g_{metal}\cdot min$) | Carbon Product. ($g_C/g_{metal}\cdot h$) * | Carbon Product./$r_{C0}$ |
| 750 | 1.2 | 0.99 | 0.83 | 0.3 | 0.16 | 0.52 |
| 800 | 2.9 | 1.40 | 0.49 | 1.1 | 0.64 | 0.56 |
| 850 | 5.9 | 1.49 | 0.25 | 2.3 | 0.74 | 0.32 |
| 900 | 12.6 | 1.87 | 0.15 | 1.7 | 0.53 | 0.30 |
| 950 | 11.22 | 1.29 | 0.11 | 1.6 | 0.34 | 0.21 |

* Carbon productivity after 2 h of reaction.

For both samples, an increase in the carbon productivity (or space–time carbon yield, calculated as the carbon concentration, $m_C$, at a given time divided by that reaction time) is observed, and also in the initial carbon growth rate ($r_{C0}$, measured from the initial slope of the $m_C$ vs. *time* curves in Figure 3) as the temperature passes from 750 °C to 900 °C, see Table 2. Thus, in the case of Ni-based catalyst, both the carbon productivity at 2 h and the initial reaction rate reached their maximum values, 1.87 and 12.60 $g_C/g_{metal}\cdot h$, respectively, at 900 °C. At higher temperatures, the intense deactivation suffered for the catalyst after an initial short period of time of around 5 min, see Figure 3a, caused a continuous decrease in the reaction rate. Consequently, the final productivity of the catalyst is severely decreased in these conditions, attaining a value of 1.29 $g_C/g_{metal}\cdot h$ at 950 °C.

In fact, the results shown in Figure 3 indicates that the reaction rate is continuously decreasing along the time on stream for all the conditions studied and also that the Co-based catalyst is less active than the Ni-based. Therefore, the decay of the catalyst activity occurs in all the cases and during all the time on stream. The value of carbon productivity attained at the end of one experiment of 2 h is the average of the decreasing reaction rates along the time, consequently, the carbon productivities are always lower than $r_{C0}$, see Table 2. The ratio between the carbon productivity and $r_{C0}$ can be taken as an indicative index of the severity of the deactivation suffered in a given experiment.

As regards the effect of reaction temperature, considering that both phenomena, the reaction and the deactivation, are activated processes; an increase in the temperature will augment both rates, see Figure 3c. However, the lower value carbon productivity attained at high temperatures, e.g., 950 °C for the Ni–Cu catalyst, is a consequence of the prevalence in this case of the deactivation over the main reaction, due to the higher value of the activation energy of the deactivation process [27]. This fact implies that the observed activation energy becomes apparently negative (ca. −30 kJ/mol), as a consequence of the deactivation of the catalyst which indicates the change in the slope of the line drawn in the high-temperature zone.

This fact is confirmed for both catalysts by the continuous decrease in the temperature of the dimensionless ratio "Carbon productivity/$r_{C0}$" shown in Table 2. In addition, it is also interesting to note that for both catalysts, at the higher temperatures studied, the initial reaction rate, which corresponds to a fresh (not deactivated) catalyst situation, does not follow the expected trend of an activated phenomenon, and the value obtained at 950 °C for Ni–Cu, 11.22 $g_C/g_{metal}\cdot$h is even lower than that obtained at 900 °C, see Table 2 and Figure 3c. This fact can be explained by a change in the catalyst selectivity towards the formation of a different type of carbon nanomaterial in these conditions, as can be seen in the characterization results section [27,30,46,47].

The effect of reaction temperature on the performance of Co–Cu/ArDC is qualitatively similar to the Ni–Cu catalyst, but the maximum value of carbon productivity (0.74 $g_C/g_{metal}\cdot$h) and $r_{CO}$ (2.28 $g_C/g_{metal}\cdot$h)) appears at a lower temperature, 850 °C, and therefore the transition of the selectivity to the formation of different carbonaceous products is produced at lower temperatures. In this regard, the apparent activation energies of the initial reaction rate, $r_{C0}$, are 155 kJ/mol for Ni–Cu/ArDC and 194 kJ/mol for Co–Cu/ArDC, see Figure 3c. These results obtained considering only the values of $r_{C0}$ in Table 2 below the maximum, confirm the better performance of the Ni–Cu/ArDC catalyst in this reaction. Furthermore, these values of the apparent activation energies are both lower than those obtained in this reaction with a Co–Cu/CDC catalyst in the same interval of temperatures, 221.7 kJ/mol.

The presence of Cu is always beneficial in increasing the activity and lifetime of the monometallic catalysts based on Ni or Co [25]. In comparison with the metals, Cu presents quite lower solubility of the carbon atoms, and also a lower methane dissociation rate (hydrogen atom abstraction from the adsorbed methane molecules). These facts reduce the formation of amorphous carbon deposits (i.e., deactivating carbon) explaining the increase in the net rate of reaction in presence of Cu.

On the other side, although the procedure of preparation of the catalyst is the same, the behavior of the different metals, Ni–Cu vs. Co–Cu, during the stage of thermal decomposition under a reductive atmosphere is clearly different. Thus, the results in Table 1 (Textural properties) and in Figures 1 and 2 reveal some important differences. During the thermal decomposition of the Argan shells impregnated with the metallic precursors, the process that really occurs is fast catalytic pyrolysis that decomposes both the metallic and the carbon precursors, producing an in situ pyrolysis–gasification of the evolving raw material (Argan). The different catalytic activity of Ni for the pyrolysis compared to Co, and also the ability of the Ni and Cu to form an alloy, explain the different structural properties of the carbon formed, and also the different nanoparticle size distributions presented in Figure 2. Consequently, the higher activity of Ni–Cu catalyst with respect to the Co–Cu sample can be due to the different metallic nanoparticle size distribution of these catalysts, shown on the inserts in Figure 2. Thus, the Co–Cu/ArDC sample, Figure 2b, contains a larger amount of small metal nanoparticles (<5 nm), while the quantity of the small particles is significantly lower for the Ni-based catalyst. The catalyst synthesis method used here determines the porous structure of the carbonaceous support developed and therefore the metallic particle size distribution [36,48]. If the microporosity of the support formed is very high, as in the case of the Co–Cu, a large fraction of the smaller metallic nanoparticles is confined to the internal structure of these micropores, remaining embedded inside the carbon matrix. In these conditions, the diffusion of the methane molecules to the exposed surface of the metallic nanoparticles is hindered, and consequently, the growth of the CNMs is slowed down [36]. Moreover, it was demonstrated that very small metallic NPs yield low growth rates and fast deactivation. On the contrary, if the NPs are very large, they have low activity due to the low exposed area, indicating the existence of an optimum size to maximize the catalyst activity [49]. The formation of an alloy in the case of the Ni–Cu catalyst extends the lifetime of the catalyst in this reaction, minimizing the formation of amorphous carbon deposits, which finally contribute to the decay of the catalyst activity [50–52]. The coupling

of all these factors explains the higher activity of Ni-based catalysts in comparison to the Co catalysts.

In general, the amount of carbon formed should increase with the metal loading, but as obtained here, this result also depends on the nanoparticle size distribution, the potential formation of an alloy, and the textural properties of the support. Thus, the Co-based catalyst has a higher metal loading (18% of Co + 5% of Cu, wt%) than the Ni sample (16% of Ni + 5% of Cu, wt%) and yet it is much less productive. Therefore, the effect of metal loading on catalyst productivity is clearly modified by the particle size distribution and the textural properties of the support, which are controlled during the catalyst preparation stage. Furthermore, the higher solubility of carbon atoms on the Ni–Cu nanoparticles also explains the higher productivity of this catalyst, but the lower selectivity for the formation of pristine CNTs, as in the case of the Co-based catalyst.

In terms of hydrogen productivity at 2 h, the values attained are 0.61 g $H_2$/$g_{metal}$·h with Ni–Cu/ArDC at 900 °C; and 0.25 g $H_2$/$g_{metal}$·h at 850 °C for Co–Cu/ArDC. All these results demonstrate the great potential of the argan residue in the development of proper catalytic supports for hydrogen production by the decomposition of methane.

Figure 4a,b show the Raman spectra of the samples after reaction at increasing temperatures, from 750 °C to 950 °C, using a feed-ratio $CH_4$/$H_2$ = 2 (28.6 % $CH_4$, 14.3 % $H_2$, 57.1 % $N_2$). For both catalysts, it was observed the appearance of a maximum value of $I_G$/$I_D$ ratio. In the case of Ni–Cu catalyst, this maximum is more intense and occurs at around 850 °C. The variation of this ratio in the case of the Co–Cu is less intense, and the maximum appears at around 900 °C.

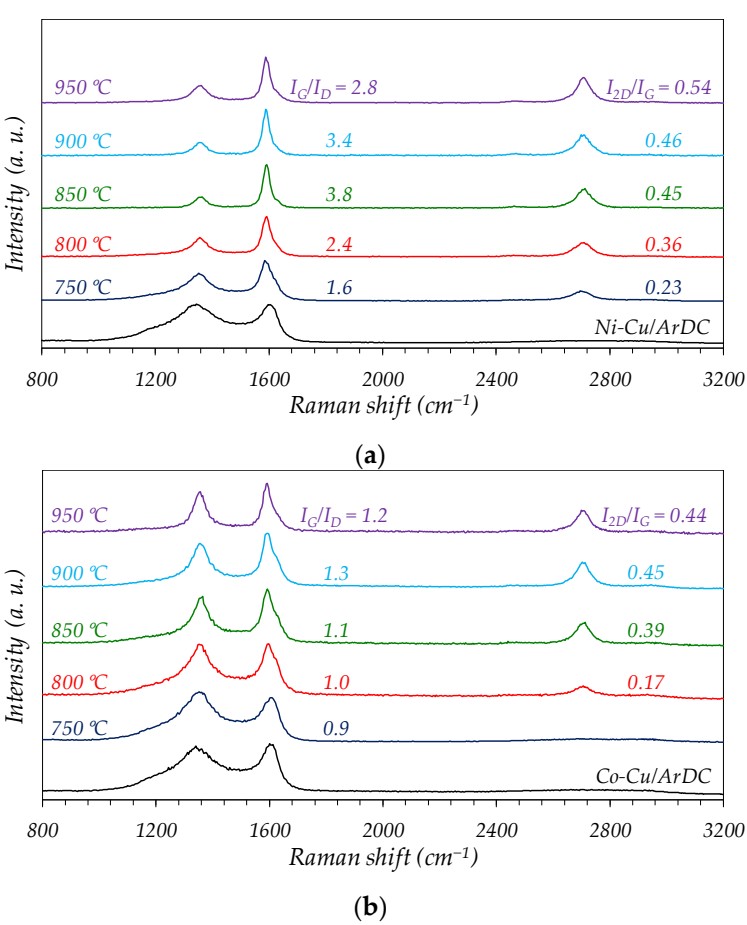

**Figure 4.** *Cont.*

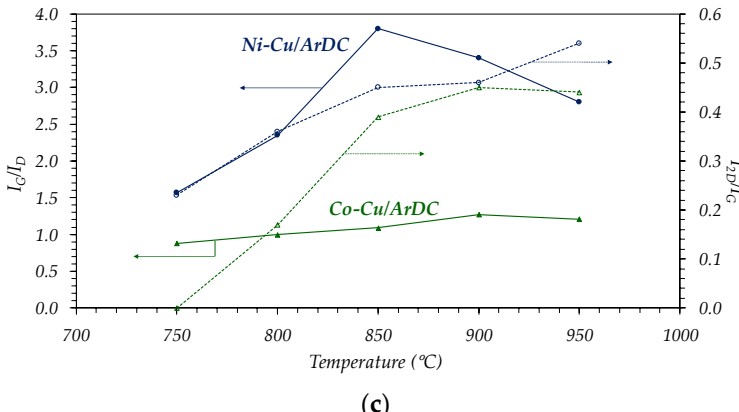

**(c)**

**Figure 4.** Influence of temperature on the Raman results of the used catalysts: (**a**) Ni–Cu/ArDC; (**b**) Co–Cu/ArDC, (**c**) $I_G/I_D$ (solid symbols) and $I_{2D}/I_G$ (open symbols) ratios.

These results are in agreement with the variation of the kinetic regime observed in Figure 3c. At low temperatures (till 850 °C), there is an increase in the intensity of the $G$ band, related to the increase in the productivity, and a decrease in the $D$ band, as a consequence of an increase in the size and crystallinity (i.e., fewer defects) of the graphitic domains of the obtained carbon material.

Above this point, the change of tendency in the evolution of the $I_G/I_D$ ratio is due to a dramatic modification of the nature of carbon formed in agreement with the kinetic results, see Figure 3, and TEM observations, see Figures 5 and 6. On the other hand, as temperature augments, the separation between the D and G bands becomes clearer, and the intermediate shoulder between both bands (corresponding to the original carbon support) almost disappeared as a result of the increase in the surface coverage by the CNMs grown, which was risen with the reaction temperature (Figure 3).

As regards the evolution of the $I_{2D}/I_G$ ratio, for the Ni–Cu/ArDC sample, the rise of this ratio suggests that the materials obtained at increasing temperatures are of nature more graphitic, with larger in-plane crystallite sizes ($L_a$) [24]. The values obtained, $I_{2D}/I_G < 0.50$, suggest that such graphitic nanostructures, which encapsulate the metallic NPs, have more than five stacked layers of graphene [53].

Figures 5 and 6 show the images of electron microscopy, SEM and TEM, of both catalysts after the reaction at the different temperatures studied. As was detected by the kinetic experiments, Figure 3, and the Raman results, Figure 4, these SEM and TEM images confirm the change in the structure and morphology observed as the temperature is increased. For the Ni–Cu/ArDC, the transition temperature is placed in the interval of 800–850 °C, and for the Co–Cu/ArDC, the change is between 800 and 850 °C. In the case of the Ni-based catalyst, Figure 5, until 800 °C the CNMs are mainly composed of CNTs and CNFs with a high proportion of defects as was also indicated by the Raman results. At temperatures above this point, the materials produced are of graphitic nature, that are encapsulating the metal nanoparticles, which are embedded inside these thick and short nanofibers. For the Co-based catalyst, Figure 6, at 750 °C the formation of pristine CNTs is still incipient, and these types of materials are formed, maintaining excellent quality, till 800 °C. From this temperature, the TEM and Raman results show a slight decrease in the quality, but the majority of products still are the CNTs. These results demonstrate that metallic catalysts supported on carbon derived from lignocellulosic residues, such as argan shells, are excellent candidates to produce CNTs of good quality, comparable to that obtained using commercial cellulose [27].

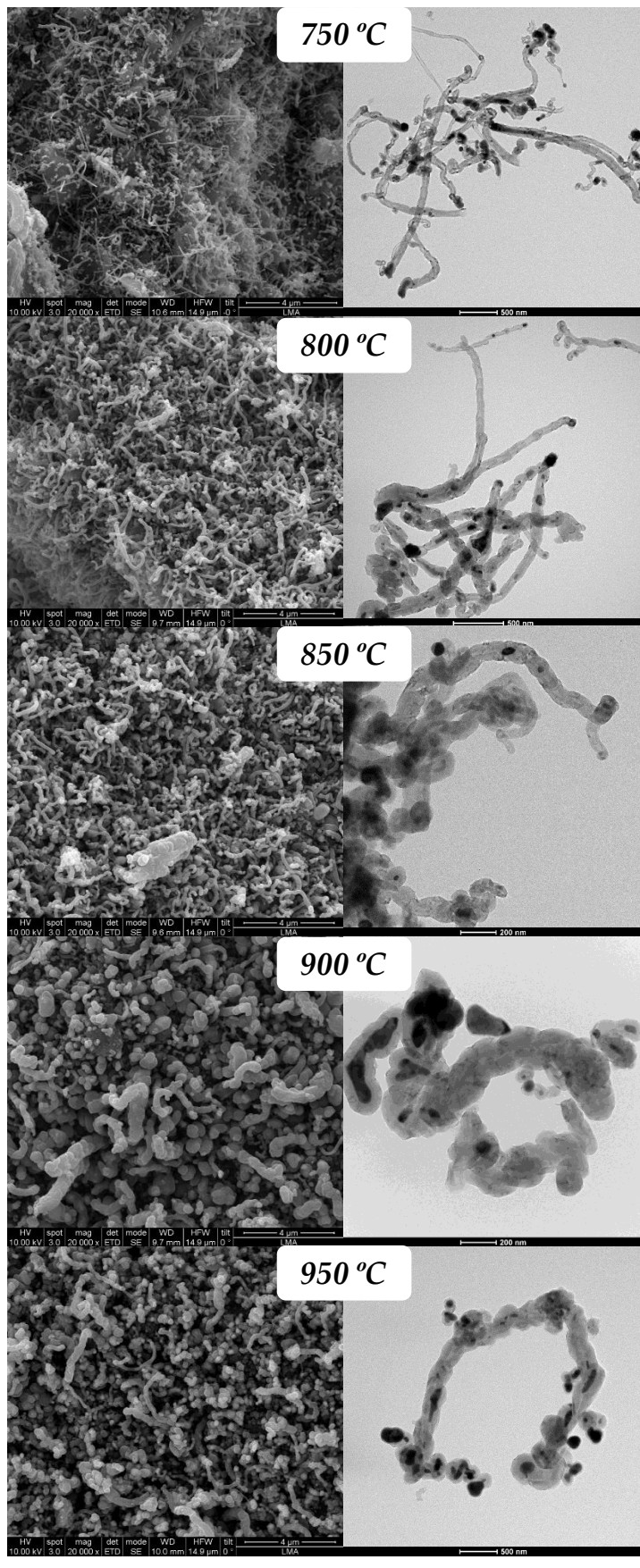

**Figure 5.** SEM and TEM images of the carbonaceous nanomaterials grown at different reaction temperatures for Ni–Cu/ArDC. Feed composition(%): $CH_4/H_2/N_2$:28.6/14.3/57.1.

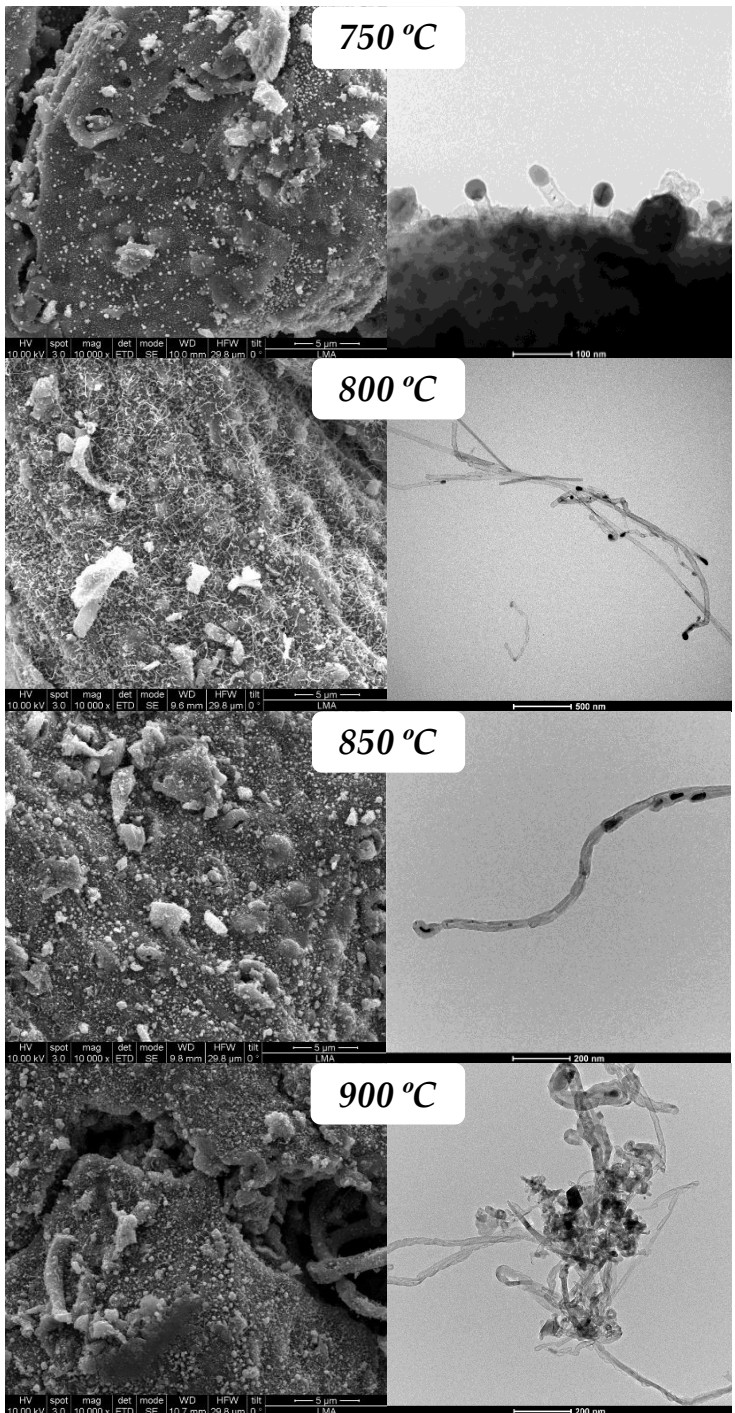

**Figure 6.** SEM and TEM images of the carbonaceous nanomaterials grown at different reaction temperatures for Co–Cu/ArDC. Feed composition (%):$CH_4/H_2/N_2$.: 28.6/14.3/57.1.

### 3.2.2. Influence of Feed Composition

The effect of the feed gas composition on the carbon production and morphology of the grown carbonaceous materials was evaluated by varying the $CH_4:H_2$ ratio from 0.5 to 3 for both catalysts. All the experiments were carried out at 800 °C given that at this temperature the selectivity towards CNTs was the highest, as just described in the previous section. Figure 7 and Table 3 show that both the reaction rate and the carbon productivity after 2 h of reaction increase with the $CH_4:H_2$ ratio in the interval studied.

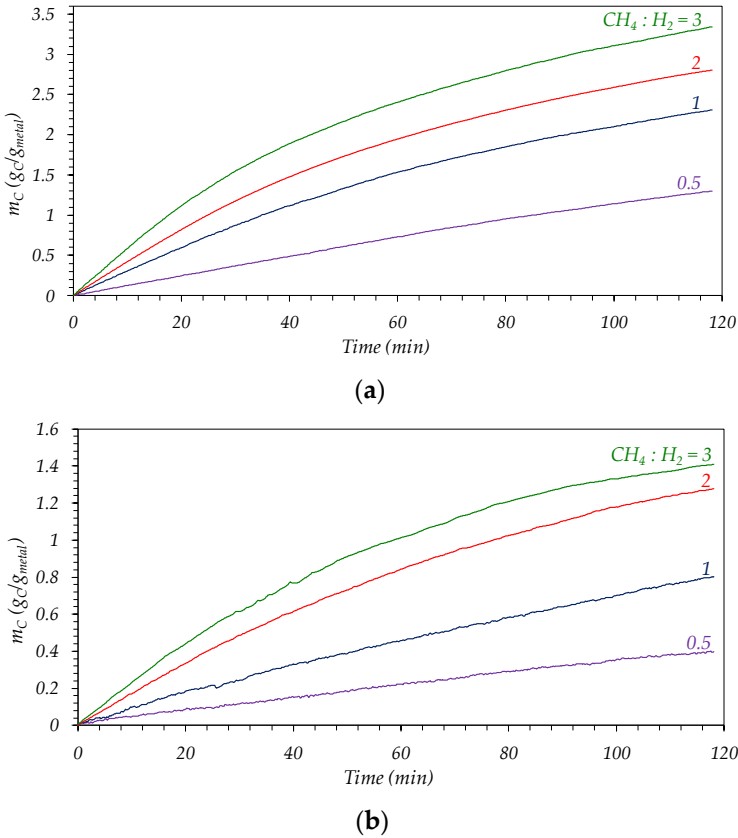

**Figure 7.** Influence of $CH_4/H_2$ feed-ratio on the evolution of carbon concentration along time: (**a**) Ni–Cu/ArDC; (**b**) Co–Cu/ArDC.

**Table 3.** Influence of $CH_4/H_2$ feed-ratio on the initial reaction rate and carbon productivity.

| CH₄:H₂ | Ni–Cu/ArDC | | | Co–Cu/ArDC | | |
|---|---|---|---|---|---|---|
| | $r_{C0}$ $(g_C/g_{metal}\cdot min)$ | Carbon Product. $(g_C/g_{metal}\cdot h)$ * | Carbon Product./$r_{C0}$ | $r_{C0}$ $(g_C/g_{metal}\cdot min)$ | Carbon Product. $(g_C/g_{metal}\cdot h)$ * | Carbon Product./$r_{C0}$ |
| 0.5 | 0.8 | 0.65 | 0.83 | 0.2 | 0.20 | 0.83 |
| 1 | 2.0 | 1.16 | 0.57 | 0.6 | 0.40 | 0.67 |
| 2 | 2.9 | 1.40 | 0.49 | 1.1 | 0.64 | 0.56 |
| 3 | 4.1 | 1.67 | 0.41 | 1.6 | 0.71 | 0.45 |

* Carbon productivity after 2 h of reaction.

As in the study of the effect of reaction temperature, the Ni-based catalyst is more active and productive that the Co catalyst at all the ratios studied. As was previously discussed, the higher proportion of small metallic NPs (<5 nm) in the Co–Cu/ArDC catalyst leaves this fraction of the metal buried in the micropores of the support, diminishing the activity of the catalyst. In addition, these small nanoparticles are less active to catalyze the formation of the CNTs. Therefore, these results give the clue for the optimization of the metallic particle size distribution of the catalyst, which is determined by the duration and the final temperature used during the thermal decomposition stage [36,48,54,55].

The maximum values of carbon productivity, 1.67 $g_C/g_{metal}\cdot h$ for Ni–Cu/ArDC and 0.71 $g_C/g_{metal}\cdot h$ for Co–Cu/ArDC, are obtained at the highest ratio CH₄:H₂, see Table 3. However, the quotient "carbon productivity/$r_{C0}$" continuously decreases as the CH₄:H₂ rises, indicating that the enhancement of both processes (carbon growth and catalyst deactivation) due to the methane content, is higher for the deactivation rate than for the carbon growth rate. That means that the apparent kinetic order with respect to methane is higher in the case of the deactivation rate.

These facts are explained considering that during the reaction of methane decomposition, the increase in methane amount in the feed enhances the carburization of the metallic nanoparticles, growing the number of carbon atoms dissolved in them and thus favoring the carbon precipitation at the metal–support interface [24,56,57]. At the same time, the presence of large amounts of carbon atoms on the exposed surface of the metallic nanoparticle, which is the situation corresponding to high methane concentrations, also favors the formation of encapsulating graphitic nanostructures that cause the catalyst deactivation. The balance between the rates of reaction and of the deactivation is quite similar for both catalysts, as is shown by the evolution of the ratio carbon productivity/$r_{C0}$, see Table 3, indicating the apparent orders of reaction for $CH_4$ and $H_2$ are similar for both catalysts.

As regards the results with the Co-based catalyst, the Raman spectra in Figure 8b are indicating the high selectivity of this catalyst for the formation of carbon nanotubes, and there are no significant differences among the spectra obtained at the different $CH_4$:$H_2$ ratios. However, for the case of $CH_4$:$H_2$ = 0.5, the Raman result is very similar to that obtained with the fresh catalyst, indicating the low coverage of the catalyst surface by the CNTs grown due to the low carbon productivity obtained under this reaction condition, see Figure 7 and Table 3.

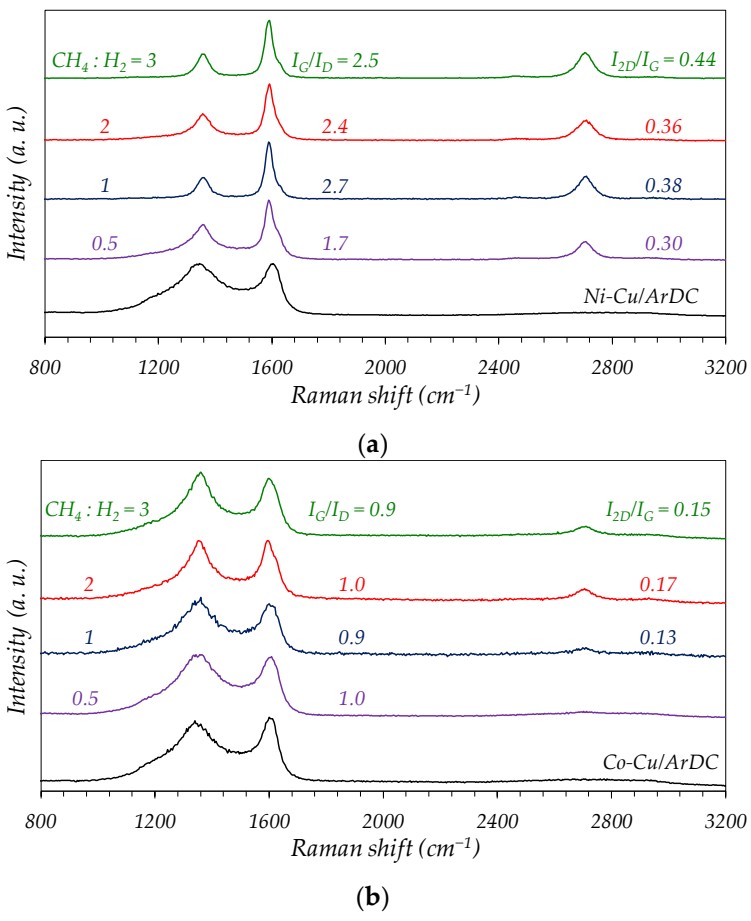

**Figure 8.** Raman spectra obtained for (**a**) Ni–Cu/ArDC and (**b**) Co–Cu/ArDC. Influence of feed composition ($CH_4$:$H_2$ ratio). Reaction temperature: 800 °C.

Figures 9 and 10 show the electron microscopy, SEM and TEM, results of the carbonaceous nanomaterials obtained at 800 °C using different $CH_4$:$H_2$ ratios for both catalysts. In the case of the Ni–Cu/ArDC, and in agreement with the kinetic results (see Figure 7a), the images in Figure 9 show that the increase in the methane concentration boosts the quantity, length and especially the thickness of the carbon materials grown, that in this case are mainly CNFs.

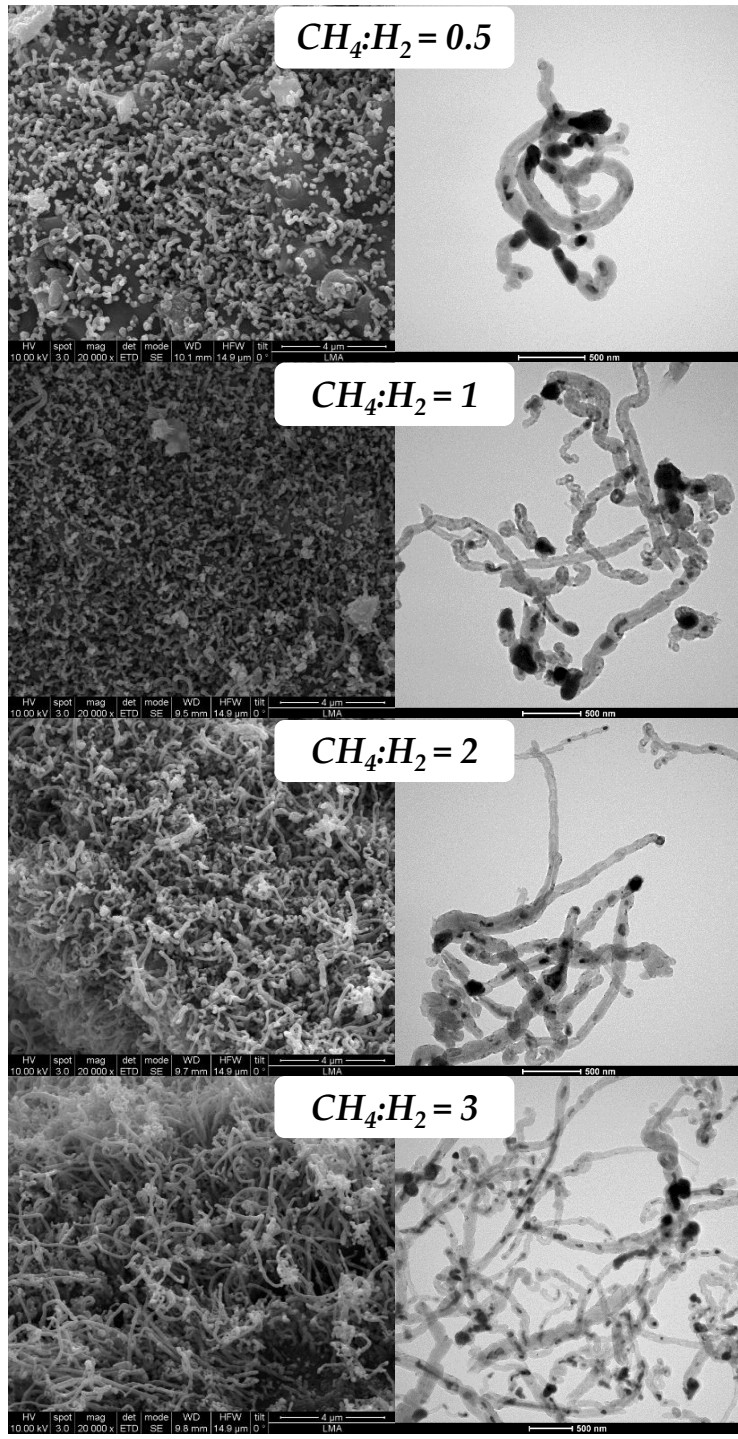

**Figure 9.** SEM and TEM images of the carbonaceous nanomaterials grown at different $CH_4:H_2$ ratios for Ni–Cu/ArDC. Reaction temperature: 800 °C.

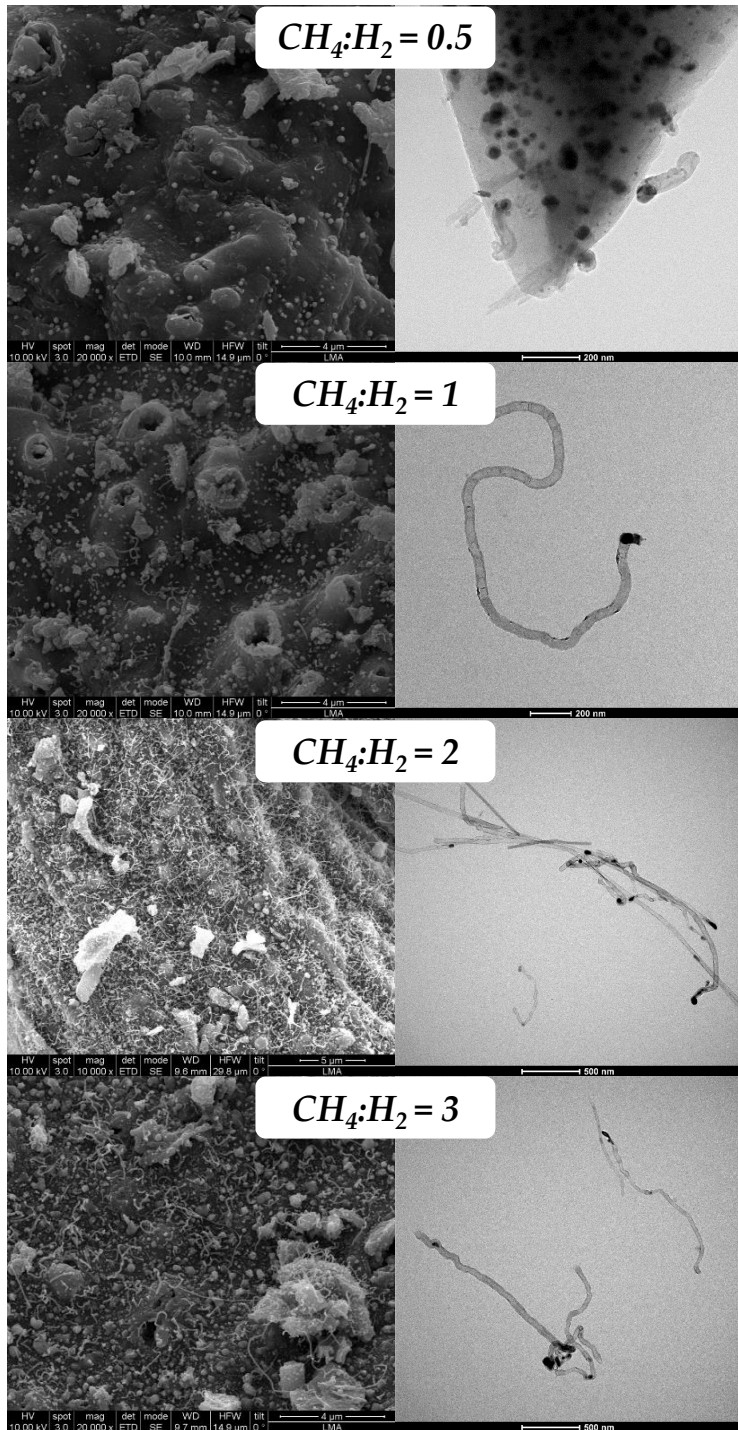

**Figure 10.** SEM and TEM images of the carbonaceous nanomaterials grown at different $CH_4:H_2$ ratios for Co–Cu/ArDC. Reaction temperature: 800 °C.

As was discussed previously, a large proportion of methane in the feed promotes the carburization step of the metallic NPs, growing the number of carbon atoms dissolved in them and thus favoring the carbon precipitation and growth. However, at the reaction temperature used (800 °C), the main part of carbon produced is in form of CNFs instead of CNTs, even at the low methane content experiment ($CH_4:H_2 = 0.5$).

In the case of Co-based catalyst, SEM and TEM results presented in Figure 10, are in total agreement with the Raman results, obtaining the selective formation of CNTs. Thus, at low methane contents ($CH_4:H_2 = 0.5$) the formation of CNTs is very incipient, see Figure 10,

in accordance with the corresponding Raman spectra in Figure 8b. In these conditions of low partial pressure of methane, the low chemical potential for the carburization of the metallic NPs slows down the diffusion and precipitation of CNTs and therefore the amount and length of the CNTs formed are very low, being only detectable by TEM, see Figure 10.

## 4. Conclusions

In this work, Ni(16%)–Cu(4%) and Co(18%)–Cu(5%) catalysts supported on carbon derived from Argan shells (ArDC) were proved to be active and selective for the production of carbonaceous nanomaterials and hydrogen via catalytic decomposition of methane. The most active catalyst was the Ni-based, reaching maximum carbon productivity of 1.87 gC/gmetal·h at 900 °C, while the best productivity of the Co-based catalyst was 0.74 gC/gmetal·h at 850 °C, in both cases with a $CH_4:H_2$ =2. These results are comparable to those obtained with catalysts supported on commercial cellulose-derived carbon.

The higher activity of the Ni catalyst is explained in terms of the different particle size distribution and the average size of the metallic NPs. Thus the Co-based catalyst presents a large proportion of small NPs, which are less active due to the effect of the diffusional restrictions caused by the presence of these smaller NPs in the micropores of the support. On the other hand, the formation of an alloy in the case of the Ni–Cu catalyst extends the lifetime of the catalyst minimizing the formation of amorphous carbon deposits, which finally cause the decay of the catalyst activity.

The kinetic results indicate that both catalysts present a maximum in the CNMs productivity with the reaction temperature. This maximum is given at around 900 °C for the catalyst based on Ni, and at 850 °C for the catalysts based on Co. In both cases, this behavior can be attributed to the fact that an increment in the reaction temperature promotes a rapid diffusion of the carbon atoms through the metal nanoparticles. This fact increases the carbon precipitation rate at the metal catalyst support interface, favoring the formation of the carbonaceous nanomaterials. At temperatures above the maximum, the deactivation of the catalyst by the formation of encapsulating graphitic structures is boosted, decreasing the carbon productivity. On the other hand, the increment in the $CH_4:H_2$ ratio produces an increase in the CNMs productivity for the two catalysts. The increase in methane partial pressure enhances the carburization of the exposed surface of the metallic nanoparticles, increasing the number of carbon atoms dissolved in them and thus favoring the carbon precipitation at the metal–support interface.

Regarding the type of carbonaceous nanomaterials obtained in the function of the operational conditions, at temperatures below 800 °C, the Ni–Cu/ArDC catalyst is selective mainly for the production of CNFs. SEM, TEM and Raman results indicate that the highest graphitic character of these CNFs was obtained at 800 °C using a $CH_4:H_2$ = 3. In the case of the Co–Cu catalyst, operating at temperatures below 850 °C, the main solid product obtained was carbon nanotubes.

The amount of carbon formed should increase with the metal loading, but as obtained here, this result also depends on the nanoparticle size distribution, the potential formation of an alloy, and the textural properties of the support. Thus, the Co-based catalyst has a higher metal loading (18% of Co + 5% of Cu, wt%) than the Ni sample (16% of Ni + 5% of Cu, wt%) and yet it is less productive. Therefore, the effect of metal loading on catalyst productivity is clearly modified by these factors, which are controlled during the catalyst preparation stage. Furthermore, the higher solubility of carbon atoms on the Ni–Cu nanoparticles also explains the higher productivity of this catalyst, but the lower selectivity for the formation of pristine CNTs, as in the case of the Co-based catalyst.

In summary, although the Co-based catalyst is less productive than the Ni-based one, their high selectivity to the CNTs formation confirms the elevated potential of the Co-based catalyst supported on carbon derived from renewable lignocellulosic residues, such as argan shells, for the production of valuable CNMs by CCVD of methane. However, in terms of productivity of hydrogen and carbon, the Ni–Cu/ArDC is the more active and therefore the more interesting for the production of $CO_x$-free hydrogen.

**Author Contributions:** Conceptualization, J.L.S., N.L., E.R., J.A. and A.M.; methodology, F.C., Z.A., M.G.-M., J.L.S., N.L., E.R., J.A. and A.M.; validation, F.C., J.L.S., N.L., E.R., J.A. and A.M.; formal analysis, J.A. and A.M.; investigation, F.C., Z.A., M.G.-M., J.L.S., N.L., E.R., J.A. and A.M.; resources, N.L., E.R., J.A. and A.M.; data curation, F.C., N.L., E.R. and A.M.; writing—original draft preparation, F.C. and A.M.; writing—review and editing, N.L., E.R., J.A. and A.M.; visualization, F.C., E.R., J.A. and A.M.; supervision, J.L.S., N.L., E.R., J.A. and A.M.; project administration, J.A. and A.M.; funding acquisition, N.L., J.A. and A.M. All authors have read and agreed to the published version of the manuscript.

**Funding:** Grant PID2020-113809RB-C31 funded by MCIN/AEI/10.13039/501100011033; grant PLEC2021-008086 funded by MCIN/AEI/10.13039/501100011033 and by the European Union NextGenerationEU/PRTR; grant PRE2018-086557 funded by MCIN/AEI/10.13039/501100011033 and by ESF Investing in your future.

**Data Availability Statement:** Not applicable.

**Acknowledgments:** The authors would like to acknowledge the use of Servicio General de Apoyo a la Investigación-SAI and Laboratorio de Microscopías Avanzadas-LMA, Universidad de Zaragoza.

**Conflicts of Interest:** The authors declare no conflict of interest.

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
