# Peer review of "Hydrogen and CNT Production by Methane Cracking Using Ni–Cu and Co–Cu Catalysts Supported on Argan-Derived Carbon"

_2305-7084, doi:10.3390/chemengineering6040047_

Round 1
Reviewer 1 Report
The authors report on H2 and CNT production by catalytic methane cracking, which is a topic of high relevance that meets the journal’s scope. The techniques applied and the overall research approach are suitable for drawing solid conclusions and in general, the manuscript deserves tob e published. However, there are a couple of issues that need to be addressed prior to publication.
· The authors are experts in the field and already published a number of high-quality studies; in the current manuscript (introduction and/or conclusion) should clearly emphasize the novelty and relevance of their current study: What is the impact of the argan-derived support, to what extent does it outperform classical formulations?
· Typically, particles sizes from 100 to 250 micrometer are chosen for kinetic testing (diffusion limiations!), however, the authors chose a particle size already starting at 10 micrometer. Could the authors please comment on why they chose such small particle sizes and how this may affect transport limitations?
· N2 was used as a balance gas; since H2 can act as an inhibitor for CH4 conversion (cf. mechanistic studies on methane pyrolysis), but N2 as an inert gas does not interfere with the CH4 decomposition pathway, the authors should provide the full range of N2 dilution (e.g. section 2.2) and should comment on the possible impact of the gas composition (inert vs. H2). This is particularly relevant in the context of Fig. 7.
· Why did the authors perform XRD for the fresh but not for the aged catalysts (cf. section 2.3, l. 125)?
· Please define the term “CDC” in Table 1 and draw a solid comparison with catalysts supported on different materials than ArDC somewhere in the manuscript (e.g. l. 319f, cellulose-based materials).
· Could the authors please briefly comment on the intersection of curves in Fig. 3 a (800°C x 950°C) and b (800°C x 900°C) in the manuscript?
· The authors should elaborate on the role of the Ni-Cu alloy; how does it contribute to a superior long-term behavior?
· Despite an overall decent language and style level, a proper proof-reading is recommended; there are several issues in the text (e.g.: l. 131: X-Rays spectroscopy => X-Ray Spectroscopy; l. 180f: the Ni-Cu/ArDC has a 63% less pore volume but its microporosity is 2.5 times higher than the Ni-Cu/ArDC one; same catalyst?!; l. 432: is less productive that the Ni-based one => is less productive than the Ni-based one; also check references [ref 2: Serp P.;, => Serp P.;]; [ ref 10: subscript]; [ref 34: onPd-Al => on Pd-Al]; [ref 28: Grunwaldt, J. => Grunwaldt, J.-D.]; [ref 31 check symbol in Effect]); figures are typically ordered in the order of appearance in the text, which is not the case in the current manuscript (e.g. ref to Fig. 4 on page 5, l. 189, with Fig. 4 on page 8; ref to Figs. 3 & 7 on page 5, l. 216, with Fig. 7 on page 10)
Author Response
Manuscript ID: ChemEngineering-1771733
Title: Hydrogen and CNTs production by methane cracking using Ni-Cu and Co-Cu catalysts supported on argan derived carbon
Authors: F. Cazaña, Z. Afailal, M. González-Martín, J. L. Sánchez, N. Latorre, E. Romeo, J. Arauzo and A. Monzón
Dear Ms. Alex Wei
We truly appreciate the detailed reviews of the submitted manuscript with reference ChemEngineering-1771733. All the comments and suggestions have been significantly useful to complete and improve the scientific discussion of the results. We have carefully checked all the comments, and the answers, text modifications and clarifications are listed below. Additionally, changes based on reviewers’ comments have been highlighted in yellow in the revised version of the manuscript file.
In the following paragraphs, a reasoned response has been added to each one of the points raised by the referees. In addition, the corresponding modifications make on the manuscript have been also highlighted.
Referees' comments:
Referee 1
The authors report on H2 and CNT production by catalytic methane cracking, which is a topic of high relevance that meets the journal’s scope. The techniques applied and the overall research approach are suitable for drawing solid conclusions and in general, the manuscript deserves to be published. However, there are a couple of issues that need to be addressed prior to publication.
1) The authors are experts in the field and already published a number of high-quality studies; in the current manuscript (introduction and/or conclusion) should clearly emphasize the novelty and relevance of their current study: What is the impact of the argan-derived support, to what extent does it outperform classical formulations?
Answer: The authors acknowledge the comment of the referee pointing the novelty of this work. In addition to the detailed study of the influence of the main operational conditions during the reaction, the real scope of this paper is related to the use of alternative and renewable materials to prepare the own catalysts. Thus, this work demonstrates that it is possible to prepare catalytic supports from lignocellulosic residua (“Biomass Derived Carbons-BDC”) without any commercial value like the “Argan Derived Carbon-ArDC” used here. The goal is to use these sustainable supports as an alternative to the traditional metallic oxides, like silica or alumina, which have a large environmental and economic impact. One of the main objectives of our research program is to apply this type of catalysts based on metals supported “Biomass Derived Carbons-BDC”, on different reactions in liquid phase (organic and aqueous) to produce fine chemicals and/or eliminate pollutants. This point is now clarified on the introduction section of the revised manuscript.
2) Typically, particles sizes from 100 to 250 micrometer are chosen for kinetic testing (diffusion limiations!), however, the authors chose a particle size already starting at 10 micrometer. Could the authors please comment on why they chose such small particle sizes and how this may affect transport limitations?
Answer: The authors acknowledge the comment of the referee. There was a typographical error in the original manuscript that has now been corrected. The true range of the particle size of the catalysts used is from 80 to 200 microns (instead 10 to 200 microns). These low values are typically selected to avoid any internal diffusion limitations, which are more severe as the size of the particles is increased. The presence of the external limitations is also minimized by using high flowrates (700 mL/min, equivalent to 1680 mL/gcat.h). This point it has been now corrected on the “Materials and Methods” section.
3) N2 was used as a balance gas; since H2 can act as an inhibitor for CH4 conversion (cf. mechanistic studies on methane pyrolysis), but N2 as an inert gas does not interfere with the CH4 decomposition pathway, the authors should provide the full range of N2 dilution (e.g. section 2.2) and should comment on the possible impact of the gas composition (inert vs. H2). This is particularly relevant in the context of Fig. 7.
Answer: The use of inert nitrogen in our case is due to an experimental requirement of the device used on the kinetic experiments (https://www.ciprecision.com/). This equipment (a thermobalance) needs to pass continuously a flow of an inert gas, N2 in our case, in order to ensure the integrity of the head of the device (the “core” of the equipment) against any potential corrosive effect of the reactive atmosphere used, which in our case is a mixture of CH4 and H2. On the paragraph 2.2 of the “Materials and Method” section is now clarified this point. On the other part, the objective of introducing hydrogen into the feed mixture is to investigate its effect on the activity and stability of the catalysts during the reaction. In absence of H2, the deactivation of the catalyst is very rapid. In fact, the experimental study allows to determine, for a given catalyst, the optimum H2/CH4 in order to minimize the deactivation, controlling also the competence of hydrogen with methane for the catalyst active sites. This point has been now clarified on the “Materials and Methods” section, and on the discussion of results.
4) Why did the authors perform XRD for the fresh but not for the aged catalysts (cf. section 2.3, l. 125)?
Answer: The authors acknowledge again this comment of the referee. The XRD patterns of the used catalysts are now included on the Figure 1 of the revised version. The results obtained indicate that the metallic phases of both catalysts (Ni-Cu and Co-Cu) do not suffer any relevant modification. However, the presence of carbon of graphitic nature after reaction is now clearly seen on the XRD patterns of the used catalysts. This point has been now clarified on the discussion of results (paragraph 3.1).
5) Please define the term “CDC” in Table 1 and draw a solid comparison with catalysts supported on different materials than ArDC somewhere in the manuscript (e.g. l. 319f, cellulose-based materials).
Answer: On the paragraph previous to Table 1 was defined the acronym CDC as “Cellulose Derived Carbon”. The comparison suggested by the referee is presented on Table 1. In order to have a relevant comparison, this is made taking the same metals and metal loadings, the same method of preparation and similar supports derived from lignocellulosic biomass.
6) Could the authors please briefly comment on the intersection of curves in Fig. 3 a (800°C x 950°C) and b (800°C x 900°C) in the manuscript?
Answer: These inserted lines at the high temperature zone correspond to the observed change of the Arrhenius tendency on the reaction rate at these high temperatures. Thus, for the Ni-Cu/ArDC catalyst, above 900 ºC, the reaction rate not only does not increase following the Arrhenius trend, but it decreases (see Table 2). This fact implies that the apparent activation energy becomes negative as a consequence of the deactivation of the catalyst as indicates the change in the slope of the line drawn in the high temperature zone. In fact, the activation energy of the deactivation process is greater than that of the CMNs growth process, and as a consequence, the observed value is negative. This same phenomenon occurs in the Co-Cu catalyst from 850 ºC. This point is clarified on the revised version of the manuscript.
7) The authors should elaborate on the role of the Ni-Cu alloy; how does it contribute to a superior long-term behaviour?
Answer: As the referee properly suggests, the possible formation of an alloy between the Ni and Cu seems to give some advantage on the performance of this catalyst in comparison with Co-Cu case. In the case of Ni-Cu, the presence of Cu improves the resistance of the catalyst against the decay of the catalyst activity, although the selectivity to the desired forms of carbon nanomaterials formed is not really enhanced.
In any case, the presence of Cu is always beneficial to increase the activity and life time of the monometallic catalysts based on Ni or Co. In comparison with Ni and Co, Cu presents quite lower solubility of carbon atoms and a lower methane dissociation rate (hydrogen atom abstraction from the adsorbed methane molecules). These facts reduce the formation of amorphous carbon deposits (i.e. deactivating carbon) explaining the increase of the net rate of reaction. Therefore, the formation of a Ni-Cu alloy could explain its higher activity. In order to really elucidate these phenomena, we are developing more deep characterization studies with the aim to optimize the catalyst composition. This point is now clarified on the discussion section (section 3.2)
8) Despite an overall decent language and style level, a proper proof-reading is recommended; there are several issues in the text (e.g.: l. 131: X-Rays spectroscopy => X-Ray Spectroscopy; l. 180f: the Ni-Cu/ArDC has a 63% less pore volume but its microporosity is 2.5 times higher than the Ni-Cu/ArDC one; same catalyst?!; l. 432: is less productive that the Ni-based one => is less productive than the Ni-based one; also check references [ref 2: Serp P.;, => Serp P.;]; [ ref 10: subscript]; [ref 34: onPd-Al => on Pd-Al]; [ref 28: Grunwaldt, J. => Grunwaldt, J.-D.]; [ref 31 check symbol in Effect]); figures are typically ordered in the order of appearance in the text, which is not the case in the current manuscript (e.g. ref to Fig. 4 on page 5, l. 189, with Fig. 4 on page 8; ref to Figs. 3 & 7 on page 5, l. 216, with Fig. 7 on page 10)
Answer: The authors deeply acknowledge all the suggestions made. The English usage of the revised version has been profoundly revised and all the mistakes have been corrected.

Reviewer 2 Report
In this manuscript, are present the results of use of Ni-Cu and Co-Cu catalysts supported 77 on Argan Derived Carbon (ArDC) on the reaction of catalytic decomposition of methane 78 to produce COx-free hydrogen and carbon nanomaterials.
The topic of manuscript is important for field of chemical engineering processes. It has enough scientifically explained experimental data. I appreciate the way catalysts are prepared, tested and characterized.
I appreciate the algorithm for testing and analyzing catalysts. The parameters that can influence the kinetics of the process are studied exactly. Scientific arguments about the characteristics of catalysts facilitate the understanding of the catalytic process.
Authors should check the correct spelling of the bibliography (there are some errors).
Author Response
Manuscript ID: ChemEngineering-1771733
Title: Hydrogen and CNTs production by methane cracking using Ni-Cu and Co-Cu catalysts supported on argan derived carbon
Authors: F. Cazaña, Z. Afailal, M. González-Martín, J. L. Sánchez, N. Latorre, E. Romeo, J. Arauzo and A. Monzón
Dear Ms. Alex Wei
We truly appreciate the detailed reviews of the submitted manuscript with reference ChemEngineering-1771733. All the comments and suggestions have been significantly useful to complete and improve the scientific discussion of the results. We have carefully checked all the comments, and the answers, text modifications and clarifications are listed below. Additionally, changes based on reviewers’ comments have been highlighted in yellow in the revised version of the manuscript file.
In the following paragraphs, a reasoned response has been added to each one of the points raised by the referees. In addition, the corresponding modifications make on the manuscript have been also highlighted.
Referees' comments:
Referee 2
In this manuscript, are present the results of use of Ni-Cu and Co-Cu catalysts supported 77 on Argan Derived Carbon (ArDC) on the reaction of catalytic decomposition of methane 78 to produce COx-free hydrogen and carbon nanomaterials.
The topic of manuscript is important for field of chemical engineering processes. It has enough scientifically explained experimental data. I appreciate the way catalysts are prepared, tested and characterized.
I appreciate the algorithm for testing and analysing catalysts. The parameters that can influence the kinetics of the process are studied exactly. Scientific arguments about the characteristics of catalysts facilitate the understanding of the catalytic process.
Authors should check the correct spelling of the bibliography (there are some errors).
Answer: The authors truly appreciate the kind comments of the referee. As noted above, the use of English and the detected mistakes on the text and on the bibliography have been revised and corrected.

Reviewer 3 Report
see attached

Author Response
Manuscript ID: ChemEngineering-1771733
Title: Hydrogen and CNTs production by methane cracking using Ni-Cu and Co-Cu catalysts supported on argan derived carbon
Authors: F. Cazaña, Z. Afailal, M. González-Martín, J. L. Sánchez, N. Latorre, E. Romeo, J. Arauzo and A. Monzón
Dear Ms. Alex Wei
We truly appreciate the detailed reviews of the submitted manuscript with reference ChemEngineering-1771733. All the comments and suggestions have been significantly useful to complete and improve the scientific discussion of the results. We have carefully checked all the comments, and the answers, text modifications and clarifications are listed below. Additionally, changes based on reviewers’ comments have been highlighted in yellow in the revised version of the manuscript file.
In the following paragraphs, a reasoned response has been added to each one of the points raised by the referees. In addition, the corresponding modifications make on the manuscript have been also highlighted.
Referees' comments:
Referee 3
The present work investigates co-production of hydrogen and CNTs via cracking of methane by supported Ni-Cu and Co-Cu catalysts on a biomass-derived carbon (from Argania spinose; ArDC). The authors present the results of the productivity and the quality of the carbonaceous material formed and the effect of the main operational conditions during the reaction, i.e. reaction temperature and feed composition. The results show that the Ni-Cu/ArDC catalyst is the most active producing 3.7 gC/gmetal (equivalent to an average hydrogen productivity of 0.61 g H2/gmetal∙h) after reaction for 2h at 900 ºC and CH4:H2 = 2. The Co-Cu/ArDC catalyst (1.4 gC/gmetal) gave a lower activity due to limited exposure of small metallic NPs (<5 nm), which are buried inside micropores of the carbonaceous support. Overall, the authors did a decent experimental work. However, the interpretation of their data and the evidence are weak. In addition, the comparison and figures are not well organized. Some supplementary data and convincing analysis are needed. Besides, there are some errors in English writing that needs correction. A Major Revision is requested based on the following specific comments.
The authors truly appreciate the comments of the referee. At it has been pointed before, the use of English and the detected mistakes on the text and on the bibliography have been revised and corrected.
Major Comments:
1) In Introduction, there is lack of research progress in this field and comparison with recent literatures. Please include the information.
Answer: As the referee suggests the introduction to the paper has been properly modified including the main recent progress in this field, especially those related to the use of alternative supports based on biomass derived carbon. In addition, the role of the catalyst composition on the stability and regenerability of the catalysts; and also the use of other non-conventional catalysts, e.g. natural mineral ores, has been revised. The authors acknowledge the comment.
2) Too many self-citations
Answer: Of course, in the revised version we have included some other relevant contributions that are in agreement with our previous findings in this process. Accordingly, the number of self-citations has been properly reduced.
3) If the Co-Cu is buried inside the carbon support, why is the same did not happen to Ni-Cu catalyst since the same synthesis method is applied?
Answer: This is an important point that of course has been clarified on the revised manuscript. Although the procedure of preparation of the catalyst is the same, the behavior of the different metals, Ni-Cu vs. Co-Cu, during the stage of thermal decomposition under reductive atmosphere is clearly different. Thus, the results on table 1 (Textural properties) and on Figures 1 and 2 reveal some important differences. During the thermal decomposition of the Argan shells impregnated with the metallic precursors, the process that really occurs is a fast catalytic pyrolysis that decomposes both, the metallic and the carbon precursors, producing an in-situ pyrolysis-gasification of the evolving raw material (argan). The different catalytic activity of Ni for the pyrolysis compared to Co (or eventually Fe), and also the ability of the Ni and Cu to form an alloy, explain the different structural properties of the carbon formed, and also the different nanoparticle size distributions presented on Figure 2. In this case, the smaller size of the particles obtained with the Co-based catalyst (Figure 2b) results in a large fraction of metal buried on the support formed, and therefore a lower activity for the decomposition of methane. This point has been also clarified on the revised manuscript (Paragraph 3.1).
4) Please indicate where CNTs and CNFs are located in Figure 5 and 6. Also, label clearly the dimension bar (eg., size, µm)
Answer: According to the suggestion of the referee, the size of the images and bars on Figures 5 and 6 has been enlarged. In addition, the identification of CNTs (mainly in Co-Cu catalyst) and CNFs (mainly in Fe-Cu catalyst) on these figures has been done.
5) How do you quantify the CNT formed? How the quartz microbalance works during reaction is not stated clear.
Answer: The quality of the different carbonaceous nanomaterials formed is qualitatively described by the TEM, SEM and Raman results. Thus, SEM, and specially TEM micrographs on Figures 5 and 6 show the formation of mainly CNTs on the Co-Cu sample, and CNFs on the Ni-Cu (at low temperature). In addition, the Raman ratios in Figures 4 and 8, allow to have semi-qualitative information on the amount of defects in these carbonaceous nanomaterials. Thus, the more graphitic the fibers, i.e. CNFs, the higher the ratio IG/ID. As regards the operation of the microbalance, this equipment allows, under controlled conditions of gas composition and temperature, to measure and store (in-situ), during the reaction of methane decomposition, the evolution of the mass of the sample of catalyst placed in the basket of the device. Since the decomposition of methane only produces pure hydrogen and solid carbon, the increase in mass recorded corresponds directly to the amount of carbon formed, which allows an in-situ measurement of the reaction rate. These data are represented, in terms of carbon concentration vs. time, in Figures 3 and 7.
6) The dimension (size) label in Fig.9 and 10 is not clear.
Answer: As in the case of Figures 5 and 6 the size of the pictures and bars has been now enlarged.
- The comparison of both catalysts were not compared in the same basis, i.e., the temperature and the loading. Therefore, the conclusion that Cu-Ni one is better cannot convince me.
The most active catalyst was the Ni-based, reaching a maximum carbon productivity 2 of 1.87 gC/gmetal∙h at 900 °C, while the best productivity of the Co-based catalyst was 1.74 gC/gmetal∙h at 850 °C, in both cases with a CH4:H2 =2.
Answer: Unfortunately, there was and typographical mistake in the “Conclusion” section. As it is shown in Table 2 and in Figure 3b, the higher productivity of the Co-Cu catalyst is 0.74 gC/gmetal∙h, not 1.74. This mistake has been corrected in the revised manuscript.
8) There are conflicts on Abstract and Conclusion. Please explain.
In Conclusion: “The most active catalyst was the Ni-based, reaching a maximum carbon productivity of 1.87 gC/gmetal∙h at 900 °C, while the best productivity of the Co-based catalyst was 1.74 gC/gmetal∙h at 850 °C, in both cases with a CH4:H2 =2”.
In Abstract: “The results show that the operation at 900 ºC and CH4:H2 = 2 with the Ni-Cu/ArDC catalyst is the most active, producing 3.7 gC/gmetal after 2 h of reaction (equivalent to an average hydrogen productivity of 0.61 g H2/gmetal∙h). The lower productivity of Co-Cu/ArDC catalyst (1.4 gC/gmetal).”
Answer: This confusion is due to the same mistake mentioned in the previous point. We apologize for these errors that have been now amended.
9). What is the relationship between carbon formed and loading of the of the metal?
Answer: This is another interesting point for this reaction. In general, the amount of carbon formed should increase with the metal loading, but as obtained here, this result also depends on the nanoparticle size distribution, the potential formation of an alloy, and the textural properties of the support. Thus, the Co-based catalyst has a higher metal loading (18% of Co + 5% of Cu, wt%) than the Ni sample (16% of Ni + 5% of Cu, wt%) and yet it is much less productive. Therefore, the effect of metal loading on catalyst productivity is clearly modified by the particle size distribution and the textural properties of the support, which are controlled during the catalyst preparation stage. Furthermore, the higher solubility of carbon atoms in Ni NPs also explains the higher productivity of this catalyst, but the lower selectivity to the formation of pristine CNTs, as in the case of the Co-based catalyst. This point has also been clarified in the revised manuscript (paragraph 3.2).

Round 2
Reviewer 3 Report
OK